# Observation of an acoustic octupole topological insulator

Haoran Xue [1,4], Yong Ge[2,4], Hong-Xiang Sun [2✉], Qiang Wang[1], Ding Jia[2], Yi-Jun Guan[2], Shou-Qi Yuan[2], Yidong Chong [1,3✉] & Baile Zhang [1,3✉]

Berry phase associated with energy bands in crystals can lead to quantised observables like quantised dipole polarizations in one-dimensional topological insulators. Recent theories have generalised the concept of quantised dipoles to multipoles, resulting in the discovery of multipole topological insulators which exhibit a hierarchy of multipole topology: a quantised octupole moment in a three-dimensional bulk induces quantised quadrupole moments on its two-dimensional surfaces, which in turn induce quantised dipole moments on one-dimensional hinges. Here, we report on the realisation of an octupole topological insulator in a three-dimensional acoustic metamaterial. We observe zero-dimensional topological corner states, one-dimensional gapped hinge states, two-dimensional gapped surface states, and three-dimensional gapped bulk states, representing the hierarchy of octupole, quadrupole and dipole moments. Conditions for forming a nontrivial octupole moment are demonstrated by comparisons with two different lattice configurations having trivial octupole moments. Our work establishes the multipole topology and its full hierarchy in three-dimensional geometries.

[1] Division of Physics and Applied Physics, School of Physical and Mathematical Sciences, Nanyang Technological University, Singapore 637371, Singapore. [2] Research Center of Fluid Machinery Engineering and Technology, Faculty of Science, Jiangsu University, Zhenjiang 212013, China. [3] Centre for Disruptive Photonic Technologies, Nanyang Technological University, Singapore 637371, Singapore. [4] These authors contributed equally: Haoran Xue, Yong Ge. ✉email: jsdxshx@ujs.edu.cn; yidong@ntu.edu.sg; blzhang@ntu.edu.sg

Topological phases of matter are generally formulated by quantised quantities expressed in terms of the Berry phase[1,2]. In the one-dimensional (1D) case, the quantisation of the electric dipole moment is associated with the Berry phase for a parallel transport of the ground state in momentum space[3,4], which leads to the concept of a 1D topological insulator (TI)[5]. The generalisations of this mathematical formulation have led to topological invariants such as the celebrated Chern number[6]. However, the relationship between the Berry phase and multipole moments, and whether multipole moments can give rise to topological phases, remained open questions until recent seminal works on quantised electric multipole TIs[7,8]. Take the quantised quadrupole insulator as an example. It has been shown, both theoretically and in recent experiments[9–14], that a two-dimensional (2D) quantised quadrupole TI exhibits topological states at "boundaries of boundaries": it lacks 1D topological edge states (unlike standard 2D TIs), but instead hosts topologically protected zero-dimensional (0D) corner states. This generalised bulk-boundary correspondence principle has opened the door to the pursuit of higher-order TIs[7–19]; topological corner states in both 2D[9–14,20,21] and three-dimensional (3D)[22–24] systems have been observed, arising from not only quantised multipole moments[9–14] but also from quantised dipole moments[20–24].

The unique feature of multipole TIs is the hierarchical nature of multipole topology. This is illustrated in Fig. 1a for the case of a 3D lattice with a three-level topological hierarchy. At the bottom level of the hierarchy, there exist quantised dipoles ($p_x$, $p_y$, $p_z$), similar to those in 1D TIs[5], giving rise to the corner states ($Q$) of the 3D lattice. At the second level, the quantised dipoles are induced along the 1D hinges by the existence of quantised quadrupoles ($q_{xy}$, $q_{yz}$, $q_{zx}$) in 2D sections of the lattice[9–14]. In addition, at the top level of the hierarchy, the quantised quadrupoles are generated by a nontrivial quantised octupole ($o_{xyz}$) in the 3D bulk. With real lattices (which are at most 3D), the

highest-order multipole TIs that can be implemented are octupole TIs. Such a lattice would enable an explicit three-level demonstration of the hierarchy of multipole TIs, which has, to our knowledge, never previously been achieved.

Here, we report on the observation of an octupole TI in an acoustic metamaterial. The structure consists of coupled acoustic resonators whose local resonances serve as artificial atomic orbitals, behaving much like theoretical tight-binding models[7,8]. Positive and negative inter-resonator couplings[9,10] are achieved by connecting the resonators with thin waveguides on different sides of each resonance's nodal line. By direct local acoustic measurements sweeping over all resonators of the sample, we have observed the in-gap 0D corner states, gapped 1D hinge states, gapped 2D surface states and gapped 3D bulk states, all of which arise from a nontrivial bulk octupole moment ($o_{xyz} = 1/2$). To corroborate these results, we have also investigated two alternative samples that have trivial octupole moments ($o_{xyz} = 0$). These trivial samples exhibit completely different boundary signatures from the nontrivial sample. This work hence demonstrates the hierarchical nature of multipole topology—from an octupole moment to quadrupole and dipole moments—which is a distinctive and unique feature of a 3D quantised octupole TI.

## Results

The lattice model[7,8] is shown schematically in Fig. 1b. The intra-cell couplings (left panel) and inter-cell couplings (right panel) have hoppings of $\gamma$ and $\lambda$, whereas solid and dashed lines denote positive and negative couplings, respectively. The unit cell can be regarded as two coupled unit cells of quadrupole TIs with opposite signs of couplings. Due to the negative couplings, there is a $\pi$ flux on each facet (see Supplementary Note 3 for the verification of $\pi$ flux in our system), which not only opens a spectral gap but also maintains the mirror symmetries (up to a gauge transformation). In the presence of mirror and inversion

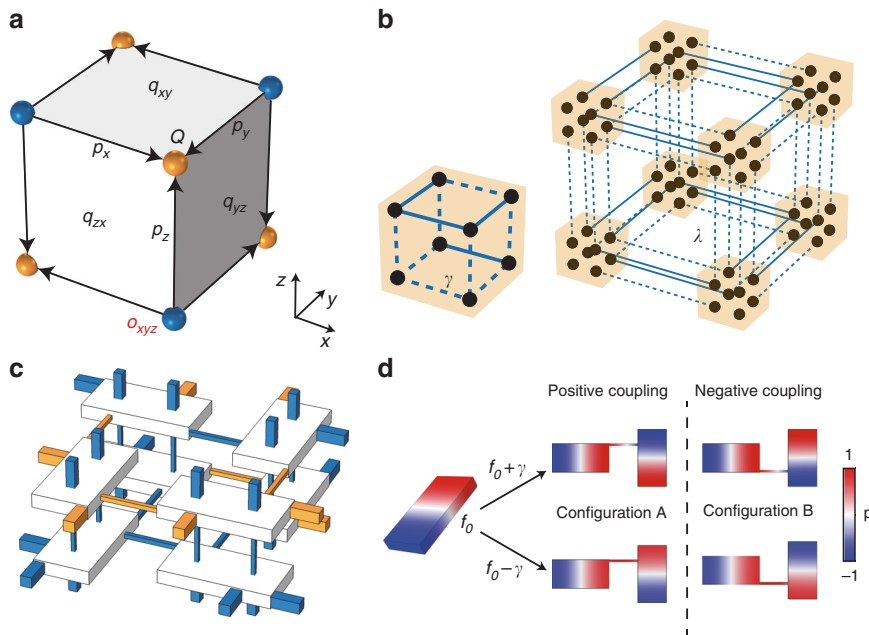

**Fig. 1 Octupole topological insulator and its acoustic metamaterial realisation. a** Schematic of a bulk octupole moment, which induces surface quadrupole moments, edge dipole moments and corner charges. **b** Tight-binding model of a quantised octupole topological insulator. The left (right) panel illustrates the intra-cell (inter-cell) couplings. Here, solid and dashed lines represent positive and negative couplings, respectively. **c** Acoustic metamaterial realisation of the model in **b**, showing one unit cell consisting of eight resonators coupled by thin waveguides (orange and blue colours denote positive and negative couplings, respectively). **d** Implementation of negative coupling with coupled acoustic resonators. The left panel shows the dipole mode of a single resonator. The right panel shows the eigenmodes of two coupled-resonator systems with coupling waveguides located at different sides of the resonance's nodal line, corresponding to opposite coupling signs.

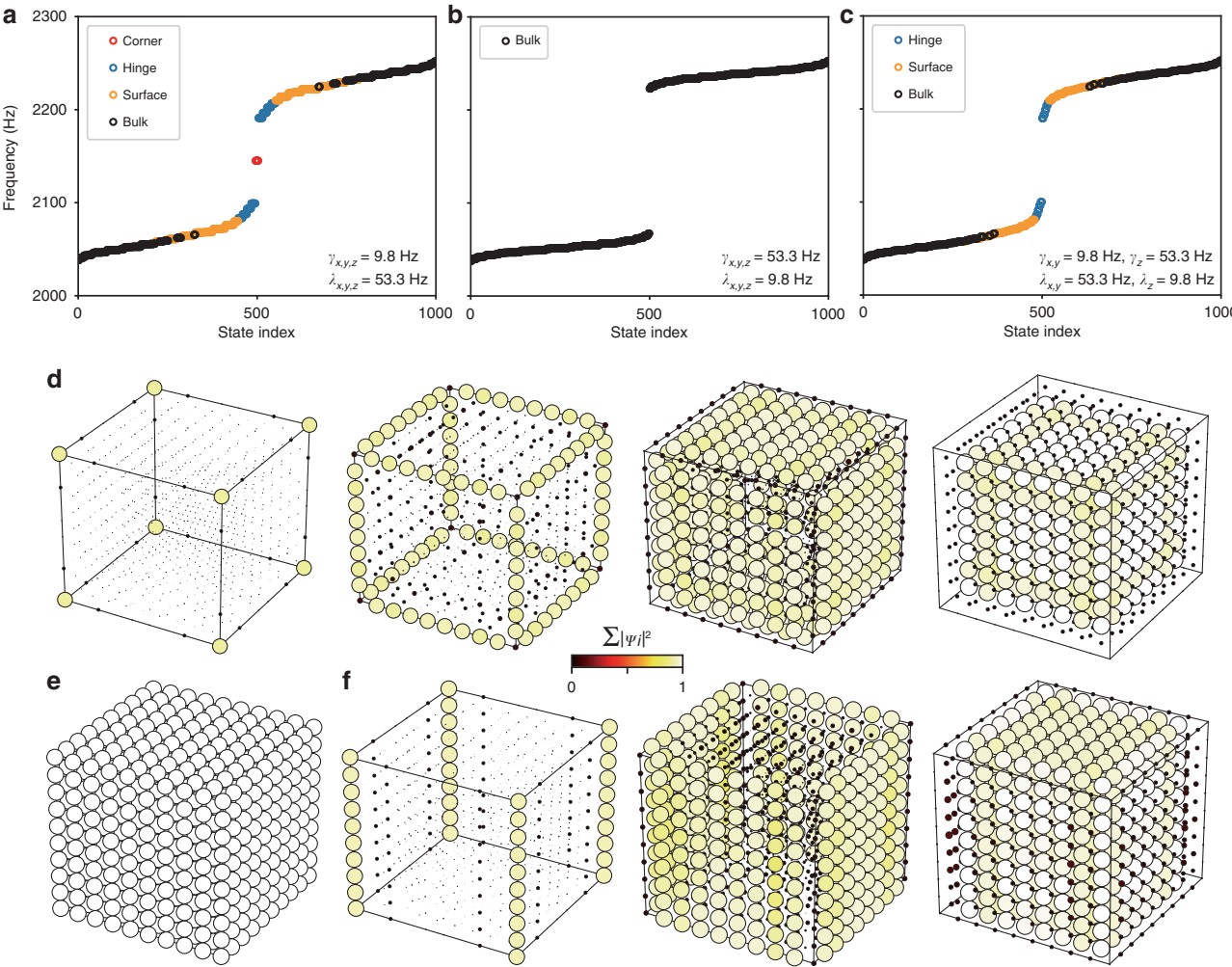

**Fig. 2 Calculations on finite lattices in topological and trivial phases. a–c** Calculated eigenvalues of 5 × 5 × 5 lattices with different coupling configurations. **d–f** Sum of probabilities for different types of states in **a–c**, respectively. In **a**, there are four types of states (bulk, surface, hinge and corner), which are plotted in **d**. In **b**, all the states are bulk states (**e**). In **c**, there are three types of states (bulk, surface and hinge), which are plotted in **f**.

symmetries, the bulk octupole moment $o_{xyz}$ takes quantised values of 0 or 1/2, depending on the relative strength of $\gamma$ and $\lambda$[7,8]. Different from the conventional TIs, the octupole TI is a so-called boundary-obstructed topological phase[25] where a topological phase transition can happen when symmetries are preserved without bulk gap closing but with some boundary gap closing.

We use coupled acoustic resonators[20–23,26–29] to implement this model (Fig. 1c). Each lattice site is a hard-walled cuboid resonator filled with air, of size 80 mm × 40 mm × 10 mm; these dimensions are chosen so that each resonator supports a dipole resonance mode (Fig. 1d, left panel) at $f_0 = 2162.5$ Hz, which is far away from other modes (Supplementary Fig. 1). Nearest-neighbour couplings are realised by connecting resonators with thin waveguides. When two resonators are coupled with a positive coupling coefficient $\gamma$, the resulting eigenmodes exhibit symmetric and antisymmetric phase relations with split eigen-frequencies $f_0 + \gamma$ and $f_0 - \gamma$. If the sign of the coupling is reversed, the eigenmodes are exchanged[30]. To achieve this, we simply relocate the connecting waveguide to the other side of the dipole mode's nodal line; the two eigenmodes then switch eigenfrequencies (Fig. 1d, right panel). The amplitude of the coupling is controlled by tuning the width of connecting wave-guide. After optimising the structure parameters to account for the resonance frequency shifts and couplings to other resonance modes (see Supplementary Note 1 for details), we arrive at the

acoustic octupole TI structure depicted in Fig. 1c, where the connecting waveguides corresponding to positive (negative) couplings are marked in orange (blue). Here, the sign of a coupling in the lattice can be determined by looking at the config-uration of associated resonators and connecting waveguide. There are only two possible in-plane configurations as shown in Fig. 1d, where the connecting waveguides are either located at the upper part (configuration A) or lower part (configuration B). Throughout this study, we assume positive (negative) coupling is implemented by configuration A (B). It is noteworthy that one can also assume positive (negative) coupling is implemented by configuration B (A), which corresponds to a gauge transformation and thus does not alter the topology of the system. Using numerical simulations and the Schrieffer–Wolff transformations[9,31], we extract the effective couplings $\gamma$ and $\lambda$, and show that the unwanted effects caused by the connecting waveguides, such as resonance frequency shifts and couplings with other modes, are negligible (Supplementary Note 1).

To demonstrate the physics of the octupole TI, three samples are constructed. Nested Wilson loops are used to reveal the topological hierarchy[7,8] (Supplementary Note 2). First, a Wilson loop along one direction shows a gapped surface spectrum. Sec-ond, a nested Wilson loop along another direction uncovers a gapped hinge spectrum. Finally, a third nested Wilson loop along the third direction gives the Wannier sector polarisation $p_\alpha^\nu$,

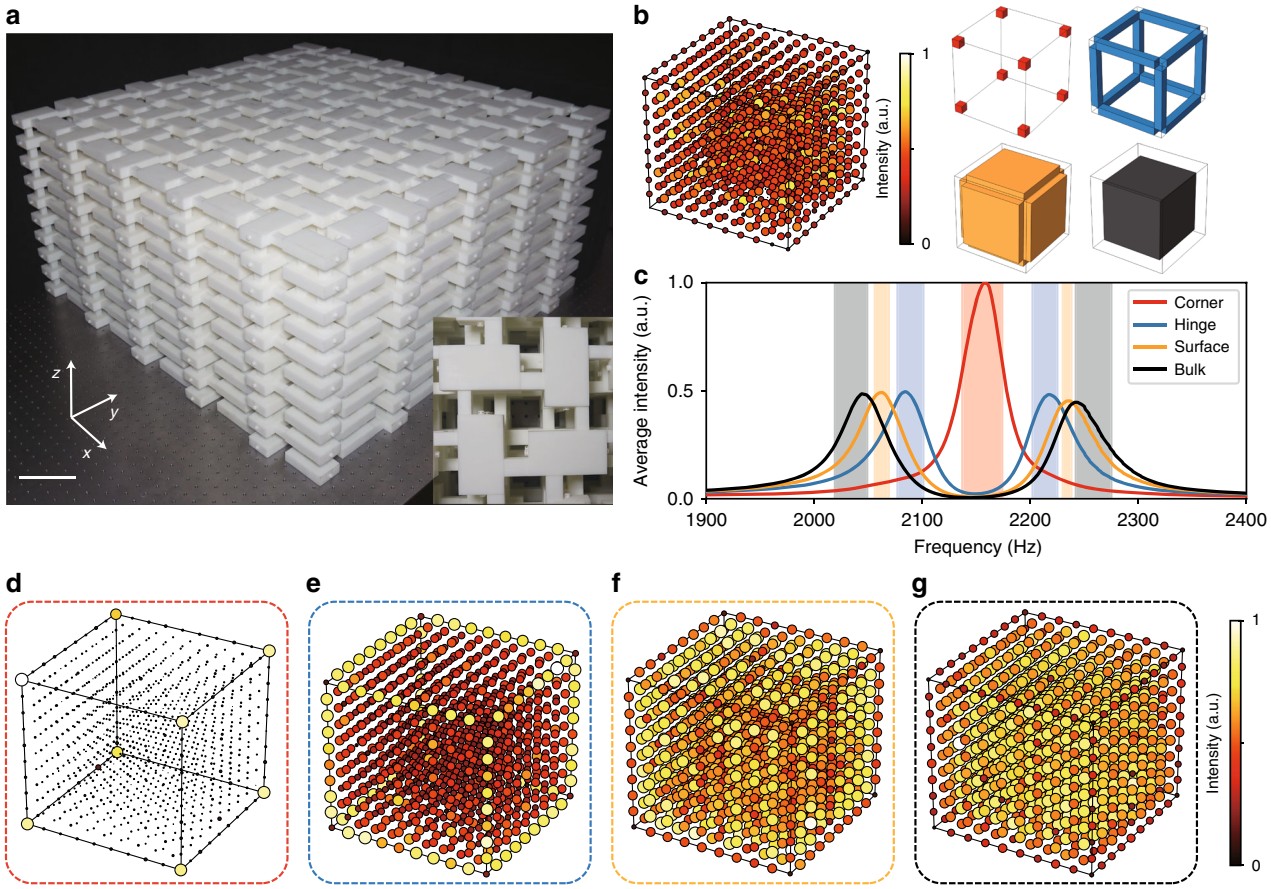

**Fig. 3 Experimental demonstration of an acoustic octupole topological insulator. a** Photo of the fabricated sample with five unit cells along each direction. The scale bar is 100 mm. **b** Left panel: measured sound intensity of all sites at an arbitrarily chosen frequency (2000 Hz). Right panel: illustration of the four regions (corner, hinge, surface and bulk) used to calculate the average intensity spectra. **c** Measured average intensity spectra for a finite lattice in the topological phase. **d–g** Spatial distributions of the acoustic response intensity, integrated over different sets of frequencies corresponding to the corner (**d**), hinge (**e**), surface (**f**) and bulk (**g**) spectra; the frequency ranges are indicated by the corresponding coloured regions in **c**.

where $v$ refers to the Wannier sector and $\alpha = x, y, z$. We calculate the Wannier sector polarisations and eigenmodes for finite lattices with parameters obtained from simulations. For the first sample, which has intra-cell couplings smaller than inter-cell couplings ($\gamma = 9.8$ Hz $< \lambda = 53.3$ Hz), the topological indices are close to $\{p_x^v, p_y^v, p_z^v\} = \{1/2, 1/2, 1/2\}$, indicating that the hinges have nontrivial dipole moments, which are induced by surface quadrupole moments, which in turn arise from the bulk octupole moment. Calculations for a finite $5 \times 5 \times 5$ lattice (which contains 1000 sites) show the physical consequences of the bulk topological properties, in the form of gapped bulk states, gapped surface states, gapped hinge states, and in-gap corner states (Fig. 2a, d).

The second sample that we fabricated has intra-cell couplings larger than inter-cell couplings ($\gamma = 53.3$ Hz $> \lambda = 9.8$ Hz). In this case, $\{p_x^v, p_y^v, p_z^v\} = \{0, 0, 0\}$, meaning that there are only bulk states and no topologically guaranteed corner states, which is consistent with numerical results (Fig. 2b, e).

The third sample is more subtle: the intra-cell couplings are smaller than the inter-cell couplings along the $x$ and $y$ directions ($\gamma_{x,y} = 9.8$ Hz $< \lambda_{x,y} = 53.3$ Hz), but the reverse is true in the $z$ direction ($\gamma_z = 53.3$ Hz $> \lambda_z = 9.8$ Hz). In this case, the topological indices are $\{p_x^v, p_y^v, p_z^v\} = \{1/2, 1/2, 0\}$. Here, the nontrivial values of $p_x^v$ and $p_y^v$ suggest each Wannier sector $v_z^{+/-}$ has a quadrupole topology ($q_{xy}^v = 1/2$) in a 2D $xy$ plane, which induces at the boundaries of the structure surface states on surfaces normal to $x$ and $y$ directions, and hinge states on hinges along $z$ (Fig. 2c, f).

However, the trivial value of $p_z^v$ indicates the induced hinge states are trivial and thus no corner states are induced.

We now probe the above signatures of the octupole TI experimentally. We measure the acoustic response to local excitations at each site of the first sample (Fig. 3a), which consists of 1000 sites. The left panel of Fig. 3b shows the measurement results at an arbitrarily chosen frequency of 2000 Hz. We divide the sample into four non-overlapping regions (Fig. 3b, right panel): the "corner" region consists of the 8 corner sites; the "hinge" region consists of the 12 hinges, each containing 8 sites; the "surface" region consists of the 6 surfaces, each containing 64 sites; and the remaining 512 sites constitute the "bulk" region. We then plot the average intensity spectra for the four regions (Fig. 3c and Methods). The peak frequencies for these spectra are found to agree well with the respective eigenfrequency ranges of the theoretically obtained corner, hinge, surface, and bulk eigenstates (Fig. 2a). Next, we plot the spatial distributions of the acoustic response integrated over frequency ranges corresponding to each type of spectral peak (the four different integration regions are indicated in Fig. 3c). The resulting distributions are indeed concentrated at the corners, hinges, surfaces and bulk of the lattice, respectively (Fig. 3d–g). The corner states are clearly observable (Fig. 3d); they exist at mid-gap frequencies distinct from the other eigenfrequencies, which are protected by chiral symmetry that approximately holds in the real structure due to the careful design process (see Supplementary Note 4 for more data on the corner state). The hinge, surface and bulk states are

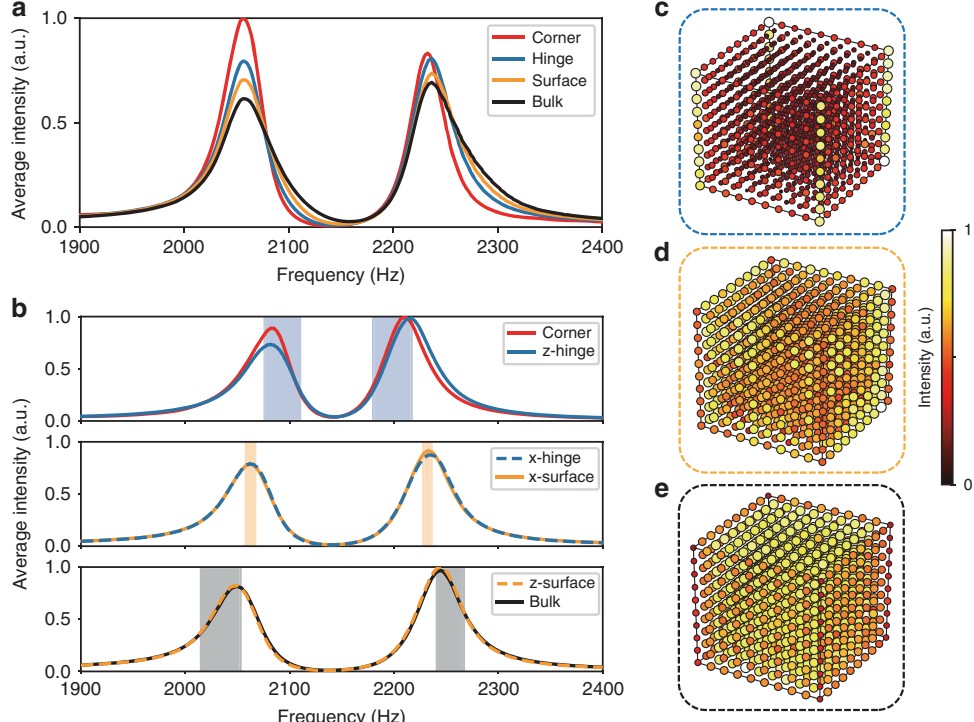

**Fig. 4 Experimental demonstration of two samples with trivial bulk octupole moment. a** Measured average intensity spectra for the sample with $\gamma > \lambda$. **b** Measured average intensity spectra for the sample with $\gamma_{x,y} < \lambda_{x,y}$ and $\gamma_z > \lambda_z$. Spectra with peaks located at similar frequencies are plotted in the same figure. Shaded regions denote the frequency regions for intensity integration. **c–e** Spatial distributions of the acoustic response intensity, integrated over the spectral peaks in **b**. These results reveal that the peaks in **b** correspond to hinge, surface and bulk states, respectively.

closer in frequency, leading to overlaps in the integrated intensity spectra (Fig. 3e–g). It is noteworthy that only the corner states are topologically protected, whereas the hinge and surface states are not.

Similar measurements were conducted on the second and third samples. For the second sample, the intensity spectra for the four different regions have identical peaks (Fig. 4a), consistent with the absence of boundary states. For the third sample, we account for the anisotropy of the lattice by treating the hinges and surfaces along different directions separately. The measured spectra in the corner region (Fig. 4b, upper panel) has peaks overlapping the spectra for the $z$ hinge region (Fig. 4c), consistent with the absence of corner states. The $x$ and $y$ hinge regions (Fig. 4b, middle panel) have spectral peaks coinciding with those of the $x$ and $y$ surface regions (Fig. 4d). (For brevity, the spectra for the $y$ hinge and $y$ surface are not plotted.) The bulk states are observed to spread into the $z$ surface region (Fig. 4b lower panel and Fig. 4e). These results match the predictions in Fig. 2 and show that the second and third samples cannot fulfil the construction of three-level hierarchy of multipole topology.

## Discussion

In conclusion, an acoustic octupole TI has been designed and demonstrated through direct local acoustic measurements. The topological in-gap corner states, gapped hinge states, gapped surface states, and gapped bulk states are all clearly distinguishable in the experimental results on the 3D lattice with nontrivial octupole moment. Contrasting experiments on different lattice configurations revealed a case where there are no boundary states at all and a case where there are hinge and surface states but no corner states, in agreement with the topological indices. These results pave the way toward studying phenomena such as topological multipole moment pumping[7,8] and the effects of non-

Hermicity[32–34], nonlinearity[35] and disorder[36] on multipole TIs, as well as the possibility of implementing reconfigurable multipole TIs with tunable corner, hinge or surface states.

**Note added**. After our submission we noticed a complimentary work[37], which also demonstrated an acoustic octupole TI.

## Methods
**Numerical and experimental details**. All numerical simulations presented in this work are performed by the commercial software COMSOL Multiphysics (Pressure Acoustics module). The photosensitive resin boundaries are modelled as sound rigid walls due to the large impedance mismatch with air ($\rho = 1.18 \text{ kg m}^{-3}$ and $\nu = 346 \text{ m s}^{-1}$).

All samples are fabricated via a stereolithography apparatus with a resin thickness of 6 mm. The samples consist of 1000 resonators (Fig. 3a in the main text) with sizes of around $1 \text{ m} \times 1 \text{ m} \times 0.5 \text{ m}$, exceeding the fabrication limit. Thus, they are divided into eight parts, which are fabricated separately and then assembled. Two small holes ($r = 2 \text{ mm}$) are located on two sides of each resonator for excitation and detection. When not in use, they are blocked with plugs.

In the experiments, the acoustic signal is launched from a balanced armature speaker, guided into the samples through a narrow tube ($r = 1.5 \text{ mm}$) and collected by a microphone (Brüel&Kjær Type 4182), which is placed at the maximum point of the dipole resonance. The measured data are processed by a Brüel&Kjær 3160-A-022 module to get the frequency spectrum.

In the measurements of site-resolved local response (Figs. 3 and 4 in the main text), the source and probe are always located at the same site. At each site $i$, we obtain the intensity of acoustic field normalised by the intensity of the source over a range of frequencies (1900–2400 Hz), denoted as $P_i(w)$. This procedure is repeated over all the 1000 sites. To eliminate the influence of varying excitation efficiencies on different sites, we normalise the data by the sum of the intensities over all frequencies to get normalised spectra: $N_i(w) = P_i(w)/\sum_w P_i(w)$. The average intensity spectra for four different regions (bulk, surface, hinge and corner) are obtained by calculating average normalised intensity within each area, respectively: $A_\alpha(w) = \sum_{i\in\alpha} N_i(w)/N_\alpha$, where $\alpha$ denotes the calculated region (i.e., bulk, surface, hinge or corner), the summation is taken over the sites within the calculated region and $N_\alpha$ is the number of sites within the calculated region.

In all figures where arbitrary units are used, the data are normalised to the maximum value in each figure.

## Data availability
The experimental data are available in the data repository for Nanyang Technological University at https://doi.org/10.21979/N9/94CPSE. Other data that support the findings of this study are available from the corresponding authors on reasonable request.

## Code availability
All numerical codes are available from the corresponding authors on reasonable request.

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

## Acknowledgements
H.X., Q.W., Y.C. and B.Z. acknowledge support from Singapore Ministry of Education under grant numbers MOE2018-T2-1-022 (S), MOE2015-T2-2-008, MOE2016-T3-1-006 and Tier 1 RG174/16 (S). Y.G., D.J., Y.-J.G., S.-Q.Y. and H.-X.S. acknowledge support from National Natural Science Foundation of China under grants 11774137 and 51779107.

## Author contributions
H.X., Y.C. and B.Z. conceived the idea. H.X. designed the sample and performed theoretical analysis. S.-Q.Y. and H.-X.S. designed the experiments. Y.G., D.J., Y.-J.G. and H.-X.S. conducted the experiments. H.X., Q.W., Y.C. and B.Z. wrote the manuscript with input from all authors. Y.C. and B.Z. supervised the whole project.

## Competing interests
The authors declare no competing interests.
