## [Peer Review File · Nature Communications]

Reviewers' comments:

Reviewer #1 (Remarks to the Author):

This is an experimental paper in which the authors designed, fabricated and characterized a 3D acoustic metamaterial which is a quantized octupole TI. They observed 0D corner states, 1D hinge states, 2D surface states, and 3D bulk states, representing the topological hierarchy of quantised octupole, quadrupole and dipole moments. This is a nice piece of work. They gave a rigorous treatment for the tight-binding model and they implemented the model with a simple and clever design that achieved negative hopping. They gave control experiments of trivial phases which makes their case more convincing. The presentation is clear and concise. I think the paper amply meets the expectation of Nature Commun.

I only find some rather minor issues which the authors might want to address.

1. According to my understanding, whether the coupling is positive or negative (as illustrated in Fig.1d) depends on the convention of the phase of the dipolar eigenstate in each cavity. There is no reason why I cannot regard the two modes in the right panel as "positive coupling", but the two coupled modes in the left panel as "negative coupling". This new convention is equivalent to the original one up to a gauge transformation. In the full structure, there are 4 cavities in each unit cell, and the cavities have different orientations. The signs of the coupling bonds are not obvious unless all the phases of the dipolar cavity modes are fixed. Therefore, I think the authors should give some explanation about their convention more explicitly, and clarify that the topology of the system does not depend on the convention of the gauge.

2. Since the fields inside the cavities have inhomogeneous distributions, it is not obvious how the acoustic intensity at each site shown in Fig. 2,3,4 is defined. Do they refer to the field intensity at a certain point in the cavity or to the average intensity inside the whole cavity?

3. The discussion of robustness of the corner states is somewhat misleading. In fact, the statements of robustness of the topological effects have sometimes been used too loosely in the field of classical topological waves. In the present case, it seems that both the two types of perturbations shown in the supplementary Fig. S4 violate the underlying symmetries (the two anti-commutative mirror symmetries) that protect the quantization of the octupole momentum. Symmetry-protected topological phases are only stable against the perturbations preserving the underlying symmetries. If the symmetry constraints are compromised, a system can continuously change from a SPT nontrivial phase to a SPT trivial phase without gap closing. Though the original Science paper, "quantized electric multipole insulators", also claimed that the corner modes are stable against disorders which breaks the mirror symmetries, there is still obvious difference between the Science paper and the present one. The Science paper used an Anderson type onsite disorder, so the mirror symmetries are preserved "on average". However, the perturbations in the present paper violate the mirror symmetries even on average.

Quantitatively, Fig.S4 b shows that the change of the frequency of the corner mode is about $30/2150=1.4\%$, which has the same order of the perturbation, which is 3%. Therefore, the result cannot be deemed as an evidence of robustness against such kind of perturbation. For Fig.S4a, I think more discussions are also needed to clarify why the frequency of corner modes are unchanged.

Reviewer #2 (Remarks to the Author):

The authors realize an acoustic metamaterial with a quantized octupole moment. The paper is well written, the data conclusively shows the expected spectroscopic characteristics of a quantized octupole insulator and thus I recommend publication of this work in Nature Communications.

I have a couple of comments:

In establishing the existence of a hierarchy of multipole moments in the present work, a more careful description is needed. For example, in the abstract, the authors say that "We directly observe 0D corner states, 1D hinge states, 2D surface states, and 3D bulk states". Putting the 0D corner states on equal footing as the hinge, surface, and bulk states is a bit misleading for the following reason: in an octupole insulator -and in the presence of chiral (sublattice) symmetry- only the 0D corner states are topologically protected. Hinge and surface modes are gapped in an octupole insulator and in principle can fuse with the bulk states without altering the octupole topology. The manuscript should correct statements in relation to this discussion in various parts of it.

I would recommend the authors to reconsider including the sentence "From this perspective, the previously-realized quadrupole TI phases are 2D projections of 3D octupole TIs". It is not clear what "projection" means in this context.

In the sentence "The unit cell can be regarded as two couple unit cells of quadrupole TIs with opposite settings", the term "opposite settings" is obscure, please clarify.

In the sentence "...meaning that there are only bulk states and no topologically guaranteed boundary states", it should say "corner states", not "boundary states", as only corner states are topologically protected.

The authors should discuss the role of chiral symmetry in the protection of the corner states. Only under chiral symmetry, a nontrivial quantized octupole moment will have mid-gap corner states. This should be particularly emphasized in this system as they seem to achieve nearly perfect chiral symmetry in the setup, which is reflected in the symmetry of the spectrum in Fig. 3c.

In the caption of Fig 1, second sentence, the description of various cases with parenthesis is confusing, as they refer to two sets of different things. Separating this sentence into two sentences will solve the confusion.

Reviewer #3 (Remarks to the Author):

The work demonstrates the realization of a model known to be a higher-order topological insulator with corner states and a quantized octupole moment in a classical setup, namely as a system of coupled acoustic resonators arranged into a 3D lattice.

I have no reason to question the quality of the experimental setup or the measurements. The material is clearly presented as well.

However, I think that what is studied experimentally is neither topological nor quantized and these essential shortcomings render the results not relevant for publication in nature communication.

Topology: The topological bulk-boundary correspondence implies that one can predict, based on a bulk topological invariant taking a nontrivial value, that there will be corner states in an open geometry (in the case at hand). However, for the model considered, this bulk-boundary-correspondence requires certain symmetries to hold (both in the bulk and in the open geometry). For instance, if as C3 rotation symmetry around the (1,1,1) axis is preserved (i.e., with respect to the axis that goes through the corner in the open geometry case), this can guarantee the bulk-boundary correspondence. However, this is not discussed in the manuscript at all. Instead, mirror symmetries are mentioned, but these rather guarantee the quantization of nested Wilson loop spectra and not the bulk-boundary correspondence. Some evidence toward the topological stability of the corner modes in the spirit discussed here is given at the end of the SI, but I think the explanation is not enough to highlight the important role played by spatial symmetry protecting the topology. Also, the C3 symmetry I refer to above is evidently broken by the physical structure of the system. Thus, it remains unclear to me what symmetry actually preserves topological bulk-boundary correspondence in the case at hand, if any.

Quantization: The electronic counter-part to this system is called a quantized octupole insulator because the octupole moment of the electronic charge distribution in the unit cell is quantized. This is a measurable observable. In the acoustic case, I do not see any corresponding quantized observable. Certainly, none has been measured in the experiment. Of course, the multiply nested Wilson loop is quantized as explained in the manuscript and SI, but what bulk observable corresponds to it? Short of a filled Fermi sea in a classical system, I do not see any. In analogy, classical realizations of Chern bands (of which there are several) are also not called realizations of quantum Hall states, because there is no quantized Hall conductivity. The measurement of some bulk topological quantity would have been desirable.

In view of this, I find the terminology "quantized..." misleading for the case at hand.

As a final small remark, I would have found a demonstration of the exponential decay of the corner modes and a comparison of the decay strength with the one expected from the ratio γ/λ useful for completeness.

I think that these points on topology and quantization are so severe and fundamental to the approach taken by the authors that I do not see how they can be overcome in a revision.

Response Letter to Reviewers

We are grateful for the constructive comments on this manuscript (NCOMMS-19-39046-T) from all the reviewers.

In the text below each of the comments from each reviewer is quoted in italics and followed by the corresponding detailed response. We have also revised the manuscript and the Supplementary Information accordingly, and these updates are highlighted in red in those files. In the text below, the references to these updates are also highlighted in red.

Reviewer #1:

Reviewer Comments:

This is an experimental paper in which the authors designed, fabricated and characterized a 3D acoustic metamaterial which is a quantized octupole TI. They observed 0D corner states, 1D hinge states, 2D surface states, and 3D bulk states, representing the topological hierarchy of quantised octupole, quadrupole and dipole moments. This is a nice piece of work. They gave a rigorous treatment for the tight-binding model and they implemented the model with a simple and clever design that achieved negative hopping. They gave control experiments of trivial phases which makes their case more convincing. The presentation is clear and concise. I think the paper amply meets the expectation of Nature Commun.

I only find some rather minor issues which the authors might want to address.

Authors Response:

We thank Reviewer #1 for the high opinions and encouraging comments.

Reviewer Comments:

1. According to my understanding, whether the coupling is positive or negative (as illustrated in Fig.1d) depends on the convention of the phase of the dipolar eigenstate in each cavity. There is no reason why I cannot regard the two modes in the right panel as "positive coupling", but the two coupled modes in the left panel as "negative coupling". This new convention is equivalent to the original one up to a gauge transformation. In the full structure, there are 4 cavities in each unit cell, and the cavities have different orientations. The signs of the coupling bonds are not obvious unless all the phases of the dipolar cavity modes are fixed. Therefore, I think the authors should give some explanation about their convention more explicitly, and clarify that the topology of the system does not depend on the convention of the gauge.

Authors Response:

We thank Reviewer #1 for pointing out this issue. We agree that whether the coupling is positive or negative depends on the convention. To make our convention more explicitly, we have revised Fig. 1d and added more explanation in the main text as: “Here the sign of a coupling in the lattice can be determined by looking at the configuration of associated resonators and connecting waveguide. There are only two possible in-plane configurations as shown in Fig. 1d where the connecting waveguides are either located at the upper part (configuration A) or lower part

(configuration B). Throughout this paper we assume positive (negative) coupling is implemented by configuration A (B). Note that one can also assume positive (negative) coupling is implemented by configuration B (A), which corresponds to a gauge transformation and thus does not alter the topology of the system.”

Reviewer Comments:

2. *Since the fields inside the cavities have inhomogeneous distributions, it is not obvious how the acoustic intensity at each site shown in Fig. 2,3,4 is defined. Do they refer to the field intensity at a certain point in the cavity or to the average intensity inside the whole cavity?*

Authors Response:

The acoustic intensity is measured at the side of each resonator where the field is at the maxima. As explained in the Methods part, there are two small holes at two side of each resonator for excitation and detection. When doing experiment, we insert the microphone into one of the holes and place it just at the resonance’s maxima to collect the signal. We have added one more sentence in the Methods part as “..... and collected by a microphone (Brüel&Kjær Type 4182) which is placed at the maximum point of the dipole resonance.” to clarify this point.

Reviewer Comments:

3. *The discussion of robustness of the corner states is somewhat misleading. In fact, the statements of robustness of the topological effects have sometimes been used too loosely in the field of classical topological waves. In the present case, it seems that both the two types of perturbations shown in the supplementary Fig. S4 violate the underlying symmetries (the two anti-commutative mirror symmetries) that protect the quantization of the octupole momentum. Symmetry-protected topological phases are only stable against the perturbations preserving the underlying symmetries. If the symmetry constraints are compromised, a system can continuously change from a SPT nontrivial phase to a SPT trivial phase without gap closing. Though the original Science paper, "quantized electric multipole insulators", also claimed that the corner modes are stable against disorders which breaks the mirror symmetries, there is still obvious difference between the Science paper and the present one.*

The Science paper used an Anderson type onsite disorder, so the mirror symmetries are preserved "on average". However, the perturbations in the present paper violate the mirror symmetries even on average.

Quantitatively, Fig.S4 b shows that the change of the frequency of the corner mode is about $30/2150=1.4\%$, which has the same order of the perturbation, which is 3% . Therefore, the result cannot be deemed as an evidence of robustness against such kind of perturbation. For Fig.S4a, I think more discussions are also needed to clarify why the frequency of corner modes are unchanged.

Authors Response:

We agree that the two types of perturbations in Fig. S4 break the mirror symmetries that protect the quantization of octupole moment. We also agree that the term ‘robustness’ is not very suitable for describing the situations here. In fact, the main purpose of section D in the Supplementary Information is to study the stability of the corner modes under local perturbations,

which is meaningful for potential applications. We have revised Supplementary Information D accordingly to remove the term “robustness” and emphasized the motivation of this section by saying “As can be seen in Figs. S4a-c, the corner states are sublattice polarized. This gives a unique stability to the corner states under some local perturbations.”

Although both perturbations break the mirror symmetries, perturbation 1 acts only on the three sites next to the corner. Since the corner state is sublattice-polarized, this perturbation would have no effects on the corner state. We have added one more sentence “As can be seen, the corner state is almost unaffected since the corner state has neglectable distribution on the perturbed sites.” in Supplementary Information D to clarify this point. In contrast, perturbation 2 acts on the corner site and thus shifts the frequency of corner state. This different responses to different kinds of perturbations could be useful in applications like sensing. To address this point, we have added one sentence in Supplementary Information D as “This unique stability of corner states under local perturbations may be useful for further applications in sensing devices.”

Reviewer #2:

Reviewer Comments:

The authors realize an acoustic metamaterial with a quantized octupole moment. The paper is well written, the data conclusively shows the expected spectroscopic characteristics of a quantized octupole insulator and thus I recommend publication of this work in Nature Communications.

Authors Response:

We thank Reviewer #2 for the recommendation.

Reviewer Comments:

I have a couple of comments:

In establishing the existence of a hierarchy of multipole moments in the present work, a more careful description is needed. For example, in the abstract, the authors say that “We directly observe 0D corner states, 1D hinge states, 2D surface states, and 3D bulk states”. Putting the 0D corner states on equal footing as the hinge, surface, and bulk states is a bit misleading for the following reason: in an octupole insulator -and in the presence of chiral (sublattice) symmetry- only the 0D corner states are topologically protected. Hinge and surface modes are gapped in an octupole insulator and in principle can fuse with the bulk states without altering the octupole topology. The manuscript should correct statements in relation to this discussion in various parts of it.

Authors Response:

We thank Reviewer #2 for the insightful comments and agree that putting corner states together with hinge, surface and bulk states may mislead readers to feel that the hinge and surface states are also topologically protected. To remove the potential confusion, we have revised the abstract as well as other parts where corner, hinge, surface and bulk states are put on the same footing. The term “topological” is put in front of corner states and the term “gapped” is put in front of

hinge, surface and bulk states. Besides, we have also added the sentence “**Note only the corner states are topologically protected while the hinge and surface states are not.**” in the main text.

Reviewer Comments:

I would recommend the authors to reconsider including the sentence "From this perspective, the previously-realized quadrupole TI phases are 2D projections of 3D octupole TIs". It is not clear what "projection" means in this context.

Authors Response:

We thank Reviewer #2 for the suggestion and **have removed this sentence in the revised manuscript.**

Reviewer Comments:

In the sentence "The unit cell can be regarded as two couple unit cells of quadrupole TIs with opposite settings", the term "opposite settings" is obscure, please clarify.

Authors Response:

We have changed the term “opposite settings” to “**opposite signs of couplings**”.

Reviewer Comments:

In the sentence "...meaning that there are only bulk states and no topologically guaranteed boundary states", it should say "corner states", not "boundary states", as only corner states are topologically protected.

Authors Response:

We have revised this sentence accordingly.

Reviewer Comments:

The authors should discuss the role of chiral symmetry in the protection of the corner states. Only under chiral symmetry, a nontrivial quantized octupole moment will have mid-gap corner states. This should be particularly emphasized in this system as they seem to achieve nearly perfect chiral symmetry in the setup, which is reflected in the symmetry of the spectrum in Fig. 3c.

Authors Response:

We agree here chiral symmetry plays an important role in pinning the corner states at mid-gap. In the real structure chiral symmetry is approximately presented, as can be seen from the numerical results in Fig. S1 where the next-nearest couplings are shown to be neglectable and the band structure is symmetric. Also this can be seen from experimentally measured spectra, as mentioned by reviewer. We have added the sentence “**...which are protected by chiral symmetry that approximately holds in the real structure due to the careful design process.**” in the revised manuscript to emphasize the role of chiral symmetry.

Reviewer Comments:

In the caption of Fig 1, second sentence, the description of various cases with parenthesis is confusing, as they refer to two sets of different things. Separating this sentence into two sentences will solve the confusion.

Authors Response:

We thank the reviewer for the suggestion. We have rewritten the sentence as: **The left (right) panel illustrates the intra-cell (inter-cell) couplings. Here solid and dashed lines representing positive and negative couplings, respectively.**

Reviewer #3:

Reviewer Comments:

The work demonstrates the realization of a model known to be a higher-order topological insulator with corner states and a quantized octupole moment in a classical setup, namely as a system of coupled acoustic resonators arranged into a 3D lattice.

I have no reason to question the quality of the experimental setup or the measurements. The material is clearly presented as well.

However, I think that what is studied experimentally is neither topological nor quantized and these essential shortcomings render the results not relevant for publication in nature communication.

Authors Response:

We thank Reviewer #3 for commenting that “*I have no reason to question the quality of the experimental setup or the measurements. The material is clearly presented as well.*” Below we provide more clarifications on the topology and quantization of our system, which we hope can lift the concerns from the reviewer.

Reviewer Comments:

Topology: The topological bulk-boundary correspondence implies that one can predict, based on a bulk topological invariant taking a nontrivial value, that there will be corner states in an open geometry (in the case at hand). However, for the model considered, this bulk-boundary-correspondence requires certain symmetries to hold (both in the bulk and in the open geometry). For instance, if a C_3 rotation symmetry around the $(1,1,1)$ axis is preserved (i.e., with respect to the axis that goes through the corner in the open geometry case), this can guarantee the bulk-boundary correspondence. However, this is not discussed in the manuscript at all. Instead, mirror symmetries are mentioned, but these rather guarantee the quantization of nested Wilson loop spectra and not the bulk-boundary correspondence. Some evidence toward the topological stability of the corner modes in the spirit discussed here is given at the end of the SI, but I think the explanation is not enough to highlight the important role played by spatial symmetry protecting the topology. Also, the C_3 symmetry I refer to above is evidently broken by the physical structure of the system. Thus, it remains unclear to me what symmetry actually preserves topological bulk-boundary correspondence in the case at hand, if any.

Authors Response:

The reviewer's concern is that “*it remains unclear... what symmetry actually preserves topological bulk-boundary correspondence...*”, since “*mirror symmetries [mentioned in our work]... rather guarantee the quantization of nested Wilson loop spectra and not the bulk-boundary correspondence.*”

Actually, the mirror symmetries not only guarantee the quantization of nested Wilson loop spectra, but also lead to the quantization of the bulk octupole moment. This has been discussed in the seminal theoretical works by Benalcazar, Bernevig and Hughes (BBH) in the following papers.

Paper 1. “Quantized electric multipole insulators”, *Science* **357**, 61 (2017).

It is this *Science* paper that proposed the model of quantized octupole moment. Note that in the second to last paragraph, this *Science* paper has stressed that “*...the quantized moment is protected by the presence of all three reflection symmetries and inversion symmetry*”.

Paper 2. “Electric multipole moments, topological multipole moment pumping, and chiral hinge states in crystalline insulators”, *PRB* **96**, 245115 (2017).

This *PRB* paper further explains the model of quantized octupole moment. On page 39 of this *PRB* paper, it is discussed that “*The octupole moment O_{xyz} is odd under each of these [mirror] symmetries. In the continuum theory, this admits only the solution $O_{xyz} = 0$, but the ambiguity in the position of the electrons due to the introduction of the lattice... also allows the solution $O_{xyz} = 1/2 \text{ mod } 1$.*”

Both Paper 1 and Paper 2 ascribed the quantized octupole moment to the mirror symmetries; we have followed this reasoning in our experimental demonstration. Note that the effective tight-binding model in our work (which is plotted in Fig. S1) is almost identical to the original model of quantized octupole moment proposed by BBH.

We guess the concern of the reviewer comes from the observation that under an open geometry, the corner states can be removed by mirror symmetric perturbations without closing bulk gap. This concern can be resolved by the recently proposed concept of “boundary-obstructed topology” (see arXiv:1908.00011, where Benalcazar and Hughes are involved), which is to characterize various higher-order topological phases. Under the mirror symmetries, the quantized bulk octupole cannot change its value unless a gap closing happens, either in the bulk or at the edges. From this perspective, the mirror symmetries can preserve the bulk-boundary correspondence.

We agree that the results in Supplementary Information D are not sufficient to highlight the symmetry protection. In fact, as mentioned in the response to Reviewer #1, the role of Supplementary Information D is to study the stability of corner states (that are sublattice polarized) under local perturbations, rather than to prove the symmetry protection. To remove the confusion, we have revised Supplementary Information D accordingly.

Reviewer Comments:

Quantization: The electronic counter-part to this system is called a quantized octupole insulator because the octupole moment of the electronic charge distribution in the unit cell is quantized. This is a measurable observable. In the acoustic case, I do not see any corresponding quantized

observable. Certainly, none has been measured in the experiment. Of course, the multiply nested Wilson loop is quantized as explained in the manuscript and SI, but what bulk observable corresponds to it? Short of a filled Fermi sea in a classical system, I do not see any. In analogy, classical realizations of Chern bands (of which there are several) are also not called realizations of quantum Hall states, because there is no quantized Hall conductivity. The measurement of some bulk topological quantity would have been desirable. In view of this, I find the terminology “quantized...” misleading for the case at hand.

Authors Response:

The reviewer’s concern is on the terminology “*quantized...*”, since “*Short of a filled Fermi sea in a classical system, [the reviewer] do not see any [quantization]*”. Actually, this terminology of “*quantized*” has already been adopted in similar works. Here we list two examples.

Paper 1. C. W. Peterson, W. A. Benalcazar, T. L. Hughes, and G. Bahl, “A quantized microwave quadrupole insulator with topologically protected corner states”, *Nature* **555**, 346 (2018).

This *Nature* paper implemented a classical microwave version of the quadrupole insulator (our work is the upgraded octupole insulator). The terminology of “*quantized*” has appeared in the title. Note that Benalcazar and Hughes are a co-authors. This means two authors of the BBH model that proposed the quantized multipole moment have acknowledged the use of the terminology of “*quantized*” in such classical wave systems.

Paper 2. X. Ni, M. Li, M. Weiner, A. Alù, and A. B. Khanikaev, “Demonstration of a quantized acoustic octupole topological insulator”, arXiv:1911.06469

This arXiv paper implemented a very similar acoustic version of the octupole topological insulator as demonstrated in our work. The terminology of “*quantized*” also appeared in the title. This is an independent and competing work from the group of Prof. Alu and Prof. Khanikaev (note that this arXiv paper was posted later than ours at arXiv:1911.06068). This means the use of this terminology is widely accepted in the community.

None of the above works, together with all other experiments of multipole topological insulators, has ever measured the bulk multipole moment. What were measured in all previous experiments are the boundary features that link to the bulk multipole moment, as did in our work. For example,

Reviewer #2 commented our work as “*the data conclusively shows the expected spectroscopic characteristics of a quantized octupole insulator*”, and Reviewer #1 commented that “*They gave control experiments of trivial phases which makes their case more convincing*”.

We have realized that the terminology of “*quantized*” in the title might have caused different interpretation among readers. We have changed the title as “**Observation of an acoustic octupole topological insulator**” in the revised manuscript.

Reviewer Comments:

As a final small remark, I would have found a demonstration of the exponential decay of the corner modes and a comparison of the decay strength with the one expected from the ratio γ/λ useful for completeness.

Authors Response:

The measured decay of corner states agrees well with the prediction from the ratio γ/λ , as shown in Fig. R2. **The demonstration of the exponential decay of the corner states has been added into Supplementary Information D in the revised version.**

Figure R2 | Decay of the corner states. **a**, Measured acoustic pressure (red dots) along x-directional hinge. The measured results agree well with the exponential decay curve (blue line) predicted from the ratio γ/λ . Here the horizontal axis refers to the site index along x-directional hinge where site “0” is the corner site. **b** and **c**, the same as **a** but for y (**b**) and z (**c**) directions. In these measurements, the speaker is fixed at the corner site and the microphone scans over all sites. Signal outside the plotting range is neglectable and thus is not shown.

Reviewer Comments:

I think that these points on topology and quantization are so severe and fundamental to the approach taken by the authors that I do not see how they can be overcome in a revision.

Authors Response:

We hope the above explanations have helped to solve the concerns of Reviewer #3.

REVIEWERS' COMMENTS:

Reviewer #1 (Remarks to the Author):

The answers to my questions and comments are clear and satisfactory. I have no more comments and I maintain my original recommendation of acceptance.

Reviewer #2 (Remarks to the Author):

All my concerns and questions have been satisfactorily address by the authors and thus I recommend the publication of this manuscript in Nature Communications.

Reviewer #3 (Remarks to the Author):

I appreciate the authors reaction to almost all of my comments, including the one on topology. In particular, one needs to invoke the notion of topological protection as introduced in arXiv:1908.00011 in order for the modes to be stable. This is a 'narrower' view, or stronger constraint, than in the earlier works, because it also requires the boundary gaps to stay open. This is not necessary for the stability of hinge modes etc. in other systems.

The only thing that I am still opposed to is that the wording "quantized" octupole insulator persists throughout the manuscript. It was just deleted in the title. For instance in the abstract, it says "Here, we report on the realization of a quantised octupole TI ...". The phase realized in this work has no quantized octupole moment according to my understanding (and the authors did not try to convince me of that). Instead, they refer to other publications that used the same terminology as inappropriately. I am not of the opinion that precedence justifies to keep using inappropriate terminology. If the authors want to keep the terminology, I the minimum addition I would ask is to clarify that this terminology, albeit established, is not actually appropriate for a classical system and why.

Response Letter to Reviewers

We are grateful for the constructive comments on this manuscript (NCOMMS-19-39046A) from all the reviewers.

In the text below each of the comments from each reviewer is quoted in italics and followed by the corresponding detailed response. We have also revised the manuscript and the Supplementary Information accordingly. In the text below, the references to these updates are highlighted in red.

Reviewer #1:

Reviewer Comments:

The answers to my questions and comments are clear and satisfactory. I have no more comments and I maintain my original recommendation of acceptance.

Authors Response:

We thank Reviewer #1 for the recommendation.

Reviewer #2:

Reviewer Comments:

All my concerns and questions have been satisfactorily address by the authors and thus I recommend the publication of this manuscript in Nature Communications.

Authors Response:

We thank Reviewer #2 for the recommendation.

Reviewer #3:

Reviewer Comments:

I appreciate the authors reaction to almost all of my comments, including the one on topology. In particular, one needs to invoke the notion of topological protection as introduced in arXiv:1908.00011 in order for the modes to be stable. This is a ‘narrower’ view, or stronger constraint, than in the earlier works, because it also requires the boundary gaps to stay open. This is not necessary for the stability of hinge modes etc. in other systems.

Authors Response:

We thank Reviewer #3 for the comments. To better illustrate the special notion of topology in this system to the readers, we have added the sentence “**Different from the conventional TIs, the octupole TI is a so-called boundary-obstructed topological phase where a topological phase transition can happen when symmetries are preserved without bulk gap closing but with some**

boundary gap closing” on page 4 of the revised manuscript and include arXiv: 1908.00011 as reference 25.

Reviewer Comments:

The only thing that I am still opposed to is that the wording “quantized” octupole insulator persists throughout the manuscript. It was just deleted in the title. For instance in the abstract, it says “Here, we report on the realization of a quantised octupole TI ...”. The phase realized in this work has no quantized octupole moment according to my understanding (and the authors did not try to convince me of that). Instead, they refer to other publications that used the same terminology as inappropriately. I am not of the opinion that precedence justifies to keep using inappropriate terminology. If the authors want to keep the terminology, I the minimum addition I would ask is to clarify that this terminology, albeit established, is not actually appropriate for a classical system and why.

Authors Response:

We have thought through this issue carefully and agree with the reviewer that the wording “quantized” is inappropriate since we provide no measurements on quantized quantities in our work. **In the revised manuscript, we have removed the wording “quantized” where it is used to describe our system.**